# CROSS-SUPERVISED OBJECT DETECTION

## ABSTRACT

After learning a new object category from image-level annotations (with no object bounding boxes), humans are remarkably good at precisely localizing those objects. However, building good object localizers (i.e., *detectors*) currently requires expensive instance-level annotations. While some work has been done on learning detectors from weakly labeled samples (with only class labels), these detectors do poorly at localization. In this work, we show how to build better object detectors from weakly labeled images of new categories by leveraging knowledge learned from fully labeled base categories. We call this learning paradigm **cross-supervised object detection**. While earlier works investigated this paradigm, they did not apply it to realistic complex images (e.g., COCO), and their performance was poor. We propose a unified framework that combines a detection head trained from instance-level annotations and a recognition head learned from image-level annotations, together with a spatial correlation module that bridges the gap between detection and recognition. These contributions enable us to better detect novel objects with image-level annotations in complex multi-object scenes such as the COCO dataset.

## 1 INTRODUCTION

Deep architectures have achieved great success in many computer vision tasks including object recognition and the closely related problem of object detection. Modern detectors, such as the Faster RCNN (Ren et al., 2015), YOLO (Redmon et al., 2016), and RetinaNet (Lin et al., 2017), use the same network backbone as popular recognition models. However, even with the same backbone architectures, detection and recognition models require different types of supervision. A good detector relies heavily on precise bounding boxes and labels for each instance (we shall refer to these as *instance-level annotations*), whereas a recognition model needs only image-level labels. Needless to say, it is more time consuming and expensive to obtain high quality bounding box annotations than class labels. As a result, current detectors are limited to a small set of categories relative to their object recognition counterparts. To address this limitation, it is natural to ask, "Is it possible to learn detectors with only class labels?" This problem is commonly referred to as weakly supervised object detection (WSOD).

Early WSOD work (Hoffman et al., 2014) showed fair performance by directly applying recognition networks to object detection. More recently, researchers have used multiple instance learning methods (Dietterich et al., 1997) to recast WSOD as a multi-label classification problem (Bilen & Vedaldi, 2016). However, these weakly supervised detectors perform poorly at localization. Most WSOD experiments have been conducted on the ILSVRC (Russakovsky et al., 2015) data set, in which images have only a single object, or on the PASCAL VOC (Everingham et al., 2010) data set, which has only 20 categories. The simplicity of these data sets limits the number and types of distractors in an image, making localization substantially easier. Learning from only class labels, it is challenging to detect objects at different scales in an image that contains many distractors. In particular, as shown in our experiments, weakly supervised object detectors do not work well in *complex* multi-object scenes, such as the COCO dataset (Lin et al., 2014).

To address this challenge, we focus on a form of learning in which the localization of classes with only object labels (weakly labeled classes) can benefit from other classes that have ground truth bounding boxes (fully labeled classes). We refer to this interesting learning paradigm as *cross-supervised object detection* (CSOD). While several works (Hoffman et al., 2014; Tang et al., 2016;

Yang et al., 2019a; Redmon & Farhadi, 2017) have explored this problem before, they still have the same limitation as the WSOD work we mentioned above. Those cross-supervised object detectors work under simplified scenarios (e.g., ILSVRC data set) where images contain single objects and are object-centered. They struggle to learn under more complex and realistic scenarios, where there are multiple objects from potentially very different classes, and objects could be small and appear anywhere in the images. In this work, we show that by doing multi-task learning on both weakly-supervised base classes and fully-supervised novel classes, our model is able to learn a good detector under the CSOD setting.

More formally, we define CSOD as follows. At training time, we are given 1) images contain objects from both base and novel classes, 2) both class labels and ground truth bounding boxes for base objects, and 3) only class labels for novel objects. Our goal is to detect novel objects. In CSOD, base classes and novel classes are disjoint. Thus, it can be seen as performing fully-supervised detection on the base classes and weakly supervised detection on the novel classes. It has similarities to both transfer learning and semi-supervised learning, since it transfer knowledge from base class to novel class and have more information about some instances than other instances. However, CSOD represents a distinct and novel paradigm for learning.

The current weakly-supervised method has several drawbacks to learn from a multi objects image. As shown in Fig. 1, a weakly supervised object detector tends to detect only the most discriminating part of novel objects instead of the whole object. Notice how only the head of the person, and not the whole body, is detected. Another issue is that the localizer for one object (e.g., the horse) may be confused by the occurrence of another object, such as the person on the horse. This example illustrates the gap between detection and recognition: without ground truth bounding boxes, the detector acts like a standard recognition model – focusing on discriminating rather than detecting.

In this paper, we explore two major mechanisms for improving on this. Our first mechanism is unifying detection and recognition. Using the same network backbone architecture, recognition and detection can be seen as image-level classification and region-level classification respectively, suggesting a strong relation between them. In particular, it suggests a shared training framework in which the same backbone is used with different heads for detection and recognition. Thus, we combine a detection head learned from ground truth bounding boxes, and a recognition head learned in a weakly supervised fashion from class labels. Unlike a traditional recognition head, our recognition head produces a class score for multiple proposals and is capable of detecting objects. The second mechanism is learning a *spatial correlation module* to reduce the gap between detection and recognition. It takes several high-confidence bounding boxes produced by the recognition head as input, and learns to regress ground truth bounding boxes. By combining these mechanisms together, our model outperforms all previous models when all novel objects are weakly labeled.

In summary, our contributions are three-fold. First, we define a new task—cross-supervised object detection, which enables us to leverage knowledge from fully labeled base categories to help learn a robust detector from novel object class labels only. Second, we propose a unified framework in which two heads are learned from class labels and detection labels respectively, along with a spatial correlation module bridging the gap between recognition and detection. Third, we significantly outperform existing methods (Zhang et al. (2018a); Tang et al. (2017; 2018)) on PASCAL VOC and COCO, suggesting that CSOD could be a promising approach for expanding object detection to a much larger number of categories.

## 2 RELATED WORK

**Weakly supervised object detection.** WSOD (Kosugi et al. (2019); Zeng et al. (2019); Yang et al. (2019b); Wan et al. (2019); Arun et al. (2019); Wan et al. (2018); Zhang et al. (2018b); Ren et al. (2020); Zhang et al. (2018c); Li et al. (2019); Gao et al. (2019b); Kosugi et al. (2019)) attempts to learn a detector with only image category labels. Most of these methods adopt the idea of Multiple Instance Learning (Dietterich et al. (1997)) to recast WSOD as a multi-label classification task. Bilen & Vedaldi (2016) propose an end-to-end network by modifying a classifier to operate at the level of image regions, serving as a region selector and a classifier simultaneously. Tang et al. (2017) and Tang et al. (2018) find that several iterations of online refinement based on the outputs of previous iterations boosts performance. Wei et al. (2018) and Diba et al. (2017) use semantic segmentation based on class activation maps (Zhou et al. (2016)) to help generate tight bounding boxes. However,

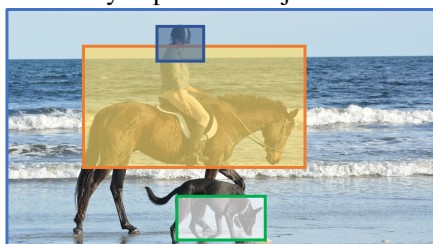      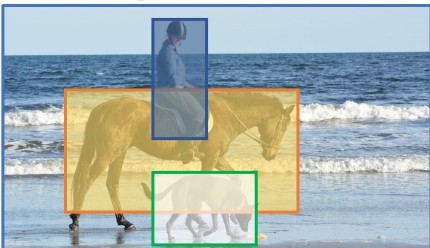

Weakly supervised object detector      Cross-supervised object detector

Figure 1: **A comparison between weakly supervised object detector and our detector.** Weakly supervised object detector only detects the most discriminating part of an object, e.g., focus on head of a person when detecting a person; or being distracted by co-occurring instances, e.g., distracted by the person on the horse when detecting a horse. Our detector can address these issues.

WSOD methods tend to focus on the most discriminating part of an object and are prone to distractions from co-occurring objects. Detecting a part of the object or distractors represents convergence to a local optimum. Thus, their performance depends heavily on initialization. In comparison, our proposed cross-supervised object detector alleviates the issue of getting trapped in a local optimum by leveraging knowledge learned from fully labeled base categories.

**Cross-supervised object detection.** There are several previous works using both image-level and instance-level annotations. Kuen et al. (2019) learned a parameter transferring function between a classifier and a detector, enabling an image-based classification network to be adapted to a region-based classification network. Hoffman et al. (2014) and Tang et al. (2016) propose methods of adaptation for knowledge transfer from classification features to detection features. Uijlings et al. (2018) use a proposal generator trained on base classes to transfer knowledge by leveraging a MIL framework, organized in a semantic hierarchy. Hoffman et al. (2015) design a three-step framework to learn a feature representation from weakly supervised classes and strongly supervised classes jointly. However, these methods can only perform object localization in single object scenes such as ILSVRC, whereas our method can perform object detection in complex multi-object scenes as well, e.g. COCO. Also, it is worth noting that we are doing multi-task learning, which means that we jointly learn from base and novel classes. In comparison, some works (Uijlings et al., 2018) are doing transfer learning. They first learn a model on base classes and then transfer and fine-tune the model on novel classes. Gao et al. (2019a) use a few instance-level labels and a large scale of image-level labels for each category in a training-mining framework, which is referred to as semi-supervised detection. Zhang et al. (2018a) propose a framework named MSD that learn objectness on base categories and use it to reject distractors when learning novel objects. In comparison, our spatial correlation module not only learns objectness, but also refines coarse bounding boxes. Further, our model learns from both base and novel classes instead of only novel classes.

## 3 CROSS-SUPERVISED OBJECT DETECTION

CSOD requires us to learn from instance-level annotations (detection labels) and image-level annotations (recognition labels). In this section, we explain the unification of detection and recognition and introduce our framework. In the next section, we describe our novel spatial correlation module.

### 3.1 UNIFYING DETECTION AND RECOGNITION

**How to learn a detector from both instance-level and image-level annotations?** Since detection and recognition can be seen as region-level and image-level classification respectively, a natural choice is to design a unified framework that combines a detection head and a recognition head that can learn from image-level and instance-level annotations respectively. Here we exploit several *baselines* to unify the detection and recognition head. (1) *Finetune*. We first learn through the detection head on base classes with fully labeled samples. Then, we finetune our model using the recognition head on novel classes with only class labels. (2) *Two Head*. We simultaneously learn the detection and recognition head on base and novel classes, respectively. The weights of the

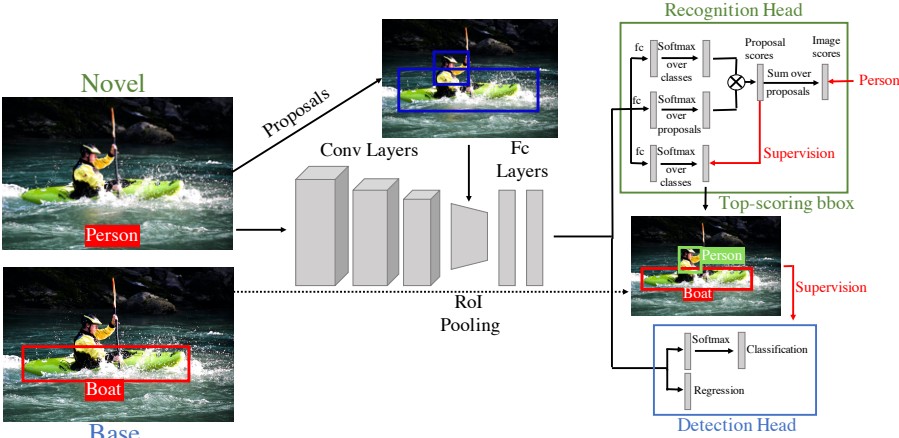

Figure 2: **Our Detection-Recognition Network (DRN) without the spatial correlation module.** In this illustration, *Person* belongs to novel classes and *Boat* belongs to base classes. The recognition head learns from the class label *Person* and outputs the top-scoring bounding box to help the detection head learn to detect the person. The spatial correlation module, discussed in § 4, can be added to further refine the top-scoring bounding boxes.

backbones are updated using the loss backpropagated from both heads jointly. (3) *Two head* $^+$. Instead of learning only on novel classes, we learn the recognition head from class labels of both base and novel classes whereas the recognition head remain the same. (4) *Two Branch.* Instead of having two shared fully-connected layers after RoI pooling layer (see Fig. 2), we make these two fully-connected layers seperated, allowing the detection and recognition head to have separate unshared pair of fully-connected layers each. Everything else is the same as the *Two Head* baseline. Experiments are conducted to compare these baselines in § 5.1 and § 5.2. Our proposed model is based on *Two Head*. We will discuss the details in § 3.2.

**The connection between the recognition and detection head.** The baselines mentioned above only use the recognition head to detect novel objects, ignoring the fact that a detection head can play the same role even better. A majority of WSOD methods (Tang et al. (2017); Wan et al. (2019); Wei et al. (2018)) find that re-train a new detector taking the top-scoring bounding boxes from a weakly supervised object detector as ground truth marginally improve the performance. Even with coarse and noisy pseudo bounding boxes, a standard object detector produces better detection results than a weakly supervised object detector. Keeping this hypothesis in mind, we introduce a guidance from the recognition head to the detection head. For each of the novel categories existing in a training sample, the recognition head outputs the top-scoring bounding box, which are then used by the detection head as supervision in that sample.

## 3.2 Detection-Recognition Network

The structure of our Detection-Recognition Network (DRN) is shown in Fig. 2. Given an image, we first generate 2000 object proposals by Selective Search (Uijlings et al. (2013)) or RPN (Ren et al. (2015)) trained on base classes. The image and proposals are fed into several convolutional (conv) layers followed by a region-of-interest (RoI) pooling layer (Girshick (2015)) to output fixed-size feature maps. Then, these feature maps are fed into two fully connected (fc) layers to produce a collection of proposal features, which are further branched into the recognition and detection head.

**Recognition Head.** We followed previous WSOD methods to design our recognition head. Since OICR (Tang et al. (2017)) is simple, neat, and commonly being used, we make our recognition head the same as OICR, but with fewer refinement branches to reduce the computation cost. However, our recognition head can be replaced by any WSOD structure as shown in § 5.3.

Within the recognition head as shown in Fig. 2, the proposal features are branched into three streams producing three matrices $\mathbf{x}^c, \mathbf{x}^d, \mathbf{x}^e \in \mathbb{R}^{C \times |R|}$, where $C$ is the number of novel classes and $|R|$ is the number of proposals. Then the two matrices $\mathbf{x}^c$ and $\mathbf{x}^d$ are passed through a softmax function

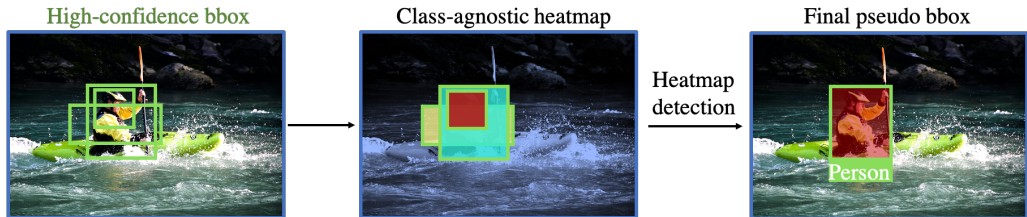

High-confidence bbox    Class-agnostic heatmap    Final pseudo bbox

Heatmap detection

Person

Figure 3: **Our spatial correlation module (SCM).** Our SCM learns to capture spatial correlation among high-confidence bounding boxes, generating a class-agnostic heatmap for the whole image. A heatmap detector is then trained to learn ground truth bounding boxes.

over classes and proposals respectively: $\sigma(\mathbf{x}^c)$ and $\sigma(\mathbf{x}^d)$. A proposal score $\mathbf{x}^R_{cr}$, indicating the score of $c^{th}$ novel class for $r^{th}$ proposal, corresponds to the respective element of the matrix $\mathbf{x}^R = \sigma(\mathbf{x}^c) \odot \sigma(\mathbf{x}^d)$, where $\odot$ refers to an element-wise product. Finally, we obtain the image score of $c^{th}$ class $\phi_c$ by summing over all proposals: $\phi_c = \sum_{r=1}^{|R|} x^R_{cr}$. Then we calculate a standard multi-class cross-entropy loss as shown in the first term of Eq.1. Another matrix $\mathbf{x}^e$ is passed through a softmax function over classes, the result of which is expresses as a weighted multi-class cross entropy loss as shown in the second term of Eq.1. We set the pseudo label for each proposal $r$ based on its IoU (or overlap) with the top-scoring proposal of $c^{th}$ class, $y_{cr} = 1$ if IoU $> 0.5$ and $y_{cr} = 0$ otherwise. The weight $w_r$ for each proposal $r$ is its IoU with the top-scoring proposal. The total loss for the recognition head is

$$L_{rec} = [-\sum_{c=1}^{C} y_c log \phi_c + (1 - y_c)log(1 - \phi_c)] + [-\frac{1}{|R|}\sum_{r=1}^{|R|}\sum_{c=1}^{C+1} w_r y_{cr} log x^e_{cr}] \qquad (1)$$

**Supervision from our recognition head.** We use the matrix $x^e$ to propose pseudos bounding boxes to guide the detection head. Specifically, we select one top-scoring proposal for each object category that appears in the image as a pseudo bounding box, as done in OICR. We introduce the spatial correlation module in § 4, to further refine this pseudo ground truth.

**Detection Head.** Now that we have pseudo bounding boxes for novel objects and ground truth bounding boxes for base objects, we train our detection head like a standard detector. For simplicity and efficiency, our detection head use the same structure of Faster R-CNN (Ren et al. (2015)). At inference time, the detection head produces detection results for both base categories and novel categories.

## 4 LEARNING TO MODEL SPATIAL CORRELATION

Our intuition is that there exists spatial correlation among high-confidence bounding boxes, and such spatial correlation can be captured to predict ground truth bounding boxes. By representing the spatial correlation in a class-agnostic heatmap, we can easily learn a mapping from recognition-based bounding boxes to ground truth bounding boxes for base categories, and then transfer this mapping to novel categories.

Thus, we propose a spatial correlation module (SCM). SCM is used as a guidance refinement technique, taking sets of high-confidence bounding boxes from the recognition head, and correspondingly returning pseudo ground truth bounding boxes to the detection head. These pseudo ground truth boxes act as supervision while training on novel categories. The framework of SCM is showed in Fig. 3. Within this module, we first generate a class agnostic heatmap based on the high-confidence bounding boxes predicted by our recognition head, and then we perform detection on top of the heatmap.

**Heatmap synthesis.** We want to capture the information about how the high-confidence bounding boxes interact amongst themselves. Here, we introduce a simple way of achieving this using a class-agnostic heatmap. For each category existing in the image $y_c = 1, c \in C$, we first threshold and select high-confidence bounding boxes of class $c$. Then we synthesize a corresponding class-agnostic heatmap, which is essentially a two-channel feature map of the same size as the original

| Method | Base mean | table | dog | horse | mbike | Novel person | plant | sheep | sofa | train | tv | **mean** |
|---|---|---|---|---|---|---|---|---|---|---|---|---|
| OICR | 42.1 | 33.4 | 29.3 | 56.3 | 64.6 | 8.0 | 23.5 | 47.2 | 47.2 | 48.3 | 61.7 | 42.0 |
| PCL | 49.2 | 51.5 | 37.3 | 63.3 | 63.9 | 15.8 | 23.6 | 48.8 | 55.3 | 61.2 | 62.1 | 48.3 |
| MSD-VGG16 | 50.6 | 14.3 | 69.3 | 65.4 | 69.6 | 2.4 | 20.5 | 54.6 | 34.3 | 58.3 | 54.6 | 44.3 |
| MSD-Ens | 53.4 | 18.3 | 70.6 | 66.7 | 69.8 | 3.7 | 24.7 | 55.0 | 37.4 | 58.3 | 57.3 | 46.1 |
| MSD-Ens+FRCN | 53.9 | 15.3 | 72.0 | 74.4 | 65.2 | 15.4 | 25.1 | 53.6 | 54.4 | 45.6 | 61.4 | 48.2 |
| Weight Transfer | 68.4 | 10.4 | 61.0 | 58.0 | 65.1 | **19.8** | 19.5 | 58.0 | 50.8 | 58.6 | 52.7 | 45.4 |
| Finetune* | 71.8 | 17.8 | 22.9 | 15.2 | 71.2 | 10.2 | 15.1 | 61.7 | 36.6 | 21.9 | 61.3 | 33.4 |
| Two Head* | **72.9** | 60.6 | 33.2 | 47.7 | 70.2 | 3.9 | 25.5 | 52.6 | 58.4 | 54.7 | 64.4 | 47.1 |
| Two Head[+]* | 72.4 | 44.5 | 29.5 | 52.4 | 68.4 | 5.1 | 22.6 | 53.0 | 55.5 | 58.6 | 64.8 | 45.4 |
| Two Branch* | 72.7 | 57.3 | 30.2 | 44.2 | 68.1 | 3.0 | 21.4 | 52.2 | 53.5 | 51.2 | 59.7 | 44.1 |
| Ours w/o SCM | 71.6 | 62.3 | 41.9 | 38.2 | **73.0** | 11.3 | 26.0 | 60.6 | **63.8** | **70.5** | 65.3 | 51.3 |
| Ours | **72.9** | 61.0 | 57.1 | 63.5 | 72.0 | 19.5 | 24.2 | 60.9 | 58.6 | 68.5 | 65.5 | **55.1**[+3.8] |
| Ours* w/o SCM | 72.7 | **66.8** | 50.4 | 57.0 | 71.5 | 12.1 | **27.6** | 57.1 | 62.7 | 54.2 | 64.2 | 52.4 |
| Ours* | 72.7 | 60.9 | **59.4** | **70.5** | 71.0 | 17.5 | 24.1 | **62.0** | 60.5 | 62.4 | **69.1** | **55.7**[+3.3] |

Table 1: **Object Detection performance (mAP %) on PASCAL VOC 2007 test set.** * indicates using the structure of OICR in the recognition head. "MSD-Ens" is the ensemble of AlexNet and VGG16. "MSD-Ens+FRCN" indicates using an ensemble model to predict pseudo ground truths and then learn a Fast-RCNN (Girshick (2015)) using VGG-16.

image. The value at each pixel is the sum and the maximum of confidence over all selected bounding boxes covering that pixel.

**Heatmap detection.** We consider each class-agnostic heatmap as a two-channel image, and perform detection on it. Specifically, we learn a class-agnostic detector on base classes, that we further use to produce pseudo ground truth bounding boxes for novel objects.

For this task, we use a lightweight one-stage detector, consisting of only five convolutional layers. We follow the same network architecture and loss as FCOS (Tian et al. (2019)), replacing the backbone and feature pyramid network with five max pooling layers. In our experiments, we also compare this tiny detector to a baseline: using three fully-connected layers to regress the ground-truth location taking the coordinates of high-confidence bounding boxes as input.

**Loss of DRN.** After introducing our SCM, we can formulate the full loss function for DRN. We use $L_{rec}$, $L_{det}$, and $L_{scm}$ to indicate the losses from our recognition head, detection head, and spatial correlation module respectively. $\lambda_{rec}$, $\lambda_{det}$, and $\lambda_{scm}$ are the regularization hyperparameters used to balance the three separate loss functions. We train our DRN using the following loss:

$$L = \lambda_{rec}L_{rec} + \lambda_{det}L_{det} + \lambda_{scm}L_{scm} \tag{2}$$

# 5 EXPERIMENTS

## 5.1 PASCAL VOC

**Setup.** PASCAL VOC 2007 and 2012 datasets contain $9,962$ and $22,531$ images respectively for 20 object classes. They are divided into train, val, and test sets. Here we follow previous work (Tang et al. (2017)) to choose the trainval set ($5,011$ images from 2007 and $11,540$ images from 2012). We divide the first 10 classes into base classes and the other 10 classes into novel classes. To evaluate our methods, we calculate mean of Average Precision (mAP) based on the PASCAL criteria, $i.e.$, IOU$>$0.5 between predicted boxes and ground truths.

**Implementation details.** All our baselines, competitors and our framework are based on VGG16 (Simonyan & Zisserman (2015)) followed most of weakly supervised object detection methods. We set $\lambda_{rec} = 1$, $\lambda_{det} = 10$, and $\lambda_{scm} = 10$. We train the whole framework for 20 epochs using SGD with a momentum of $0.9$, a weight decay of $0.0005$ and a learning rate of $0.001$, which is reduced by a factor of 10 at $14^{th}$ epoch. For a stable learning process, we don't provide supervision from recognition head to detection head in the first 9 epochs.

| method | non-voc → voc: test on B = {voc} | | | | | | sixty → twenty: test on B = {twenty} | | | | | |
|---|---|---|---|---|---|---|---|---|---|---|---|---|
| | AP | $AP_{50}$ | $AP_{75}$ | $AP_S$ | $AP_M$ | $AP_L$ | AP | $AP_{50}$ | $AP_{75}$ | $AP_S$ | $AP_M$ | $AP_L$ |
| Rec. Head | 4.0 | 15.4 | 0.9 | 1.2 | 5.7 | 5.8 | 4.7 | 16.4 | 1.3 | 1.7 | 8.0 | 6.9 |
| OICR | 4.2 | 15.7 | 1.0 | 1.3 | 5.5 | 5.9 | 4.5 | 16.6 | 1.4 | 2.0 | 8.2 | 7.1 |
| PCL | 9.2 | 19.6 | - | - | - | - | 9.2 | 19.6 | - | - | - | - |
| Weight T. | 9.3 | 26.4 | 5.7 | 5.8 | 11.7 | 12.4 | 8.7 | 25.5 | 5.5 | 5.4 | 11.5 | 11.7 |
| Finetune | 2.3 | 7.4 | 0.3 | 0.7 | 3.1 | 3.3 | 2.4 | 7.7 | 0.2 | 0.5 | 2.8 | 3.0 |
| Two Head | 11.0 | 30.2 | 6.1 | 6.2 | 15.4 | 15.4 | 11.3 | 29.5 | 5.8 | 6.3 | 14.8 | 15.0 |
| Two Head$^+$ | 9.1 | 26.7 | 5.4 | 5.5 | 12.1 | 12.3 | 9.0 | 27.1 | 5.4 | 5.7 | 11.7 | 11.6 |
| Two Branch | 9.4 | 26.6 | 5.6 | 5.7 | 12.3 | 12.4 | 8.5 | 24.4 | 4.5 | 4.3 | 11.9 | 11.9 |
| Ours w/o SCM | 12.5 | 33.6 | 6.6 | **7.3** | **19.2** | 16.4 | 12.6 | 32.3 | 7.8 | 7.0 | **19.4** | 17.4 |
| Ours | **13.9**$^{+1.4}$ | **36.2**$^{+2.6}$ | **7.7** | 6.9 | 18.8 | **19.9** | **14.0**$^{+1.4}$ | **34.5**$^{+2.2}$ | **8.9** | **7.1** | 19.2 | **20.6** |

Table 2: **The results on COCO.** We compare our method with several strong baselines in § 3.1 and competitors. Our method significantly outperforms these approaches, showing that our cross-supervised object detector is capable of detecting novel objects in complex multi-object scenes.

**Baselines and competitors.** We compare against several baselines as mentioned in § 3.1, two WSOD methods: OICR (Tang et al. (2017)) and PCL (Tang et al. (2018)), and two cross-supervised object detector: MSD (Zhang et al. (2018a)), weight transfer (Kuen et al. (2019)).

**Results.** As shown in Table 1, our method outperforms all other approaches by a large margin (over 7% relative increase in mAP on novel classes). The results are consistent with our discussion in § 3.1. We note that (1) sharing backbone for the recognition and detection head learns a more discriminative embedding for novel objects. In Table 1, Two Head* boosts the performance by 5 points as compared to only using the recognition head (OICR). (2) A supervision from recognition head to detection head exploits the full potential of a detection model. By adding the supervision (Ours* w/o SCM ), the result is improved by 5 points as compared to Two Head. (3) Our spatial correlation module successfully captures the spatial correlation between high-confidence proposals. It further boosts the performance by 3 points.

| method | non-voc→voc $AP_{50}$ on B | sixty→twenty $AP_{50}$ on B |
|---|---|---|
| max | 35.5 | 33.8 |
| sum | 36.0 | 34.0 |
| num | 31.5 | 29.5 |
| max+sum | **36.2** | **34.5** |
| max+num | 35.7 | 34.1 |
| sum+num | 35.9 | 34.2 |
| max+sum+num | 36.1 | 34.2 |

(a) **Ablation on Heatmap synthesis.** The result suggests using two-channel heatmap consists of maximum confidence and sum of confidence over proposals covering that position.

| method | | non-voc→voc $AP_{50}$ on B | sixty→twenty $AP_{50}$ on B |
|---|---|---|---|
| Fc layer | 2 layer | 31.0 | 28.7 |
| | 3 layer | 30.8 | 28.3 |
| | 4 layer | 30.5 | 28.5 |
| | R-50-FPN | **36.4** | **34.8** |
| FCOS | 4 conv | 35.8 | 33.8 |
| | 5 conv | 36.2 | 34.5 |
| | w/o SCM | 33.6 | 32.3 |

(b) **Ablation on the structure of SCM.** FCOS with 5 conv layers has nearly the best performance and very few parameters compared to a ResNet-50 backbone.

| method | non-voc→voc $AP_{50}$ on B | sixty→twenty $AP_{50}$ on B |
|---|---|---|
| WSDDN | 35.7 | 33.8 |
| OICR | **36.6** | **34.7** |
| Ours | 36.4 | 34.5 |

(c) **Ablation on the structure of the recognition head.** OICR has more refinement branches so it behaves a little better than our recognition head but takes double the computation time.

| dataset | method | base→novel $AP_{50}$ on A | base→novel $AP_{50}$ on B |
|---|---|---|---|
| PASCAL VOC | RPN | **76.2** | 46.1 |
| | SS | 72.7 | **55.7** |
| non-voc→voc | RPN | **46.3** | 36.2 |
| | SS | 42.5 | 34.5 |

(d) **Ablation on the proposal generator.** On PASCAL VOC, there are not enough categories to learn a good RPN. So, we use selective search and RPN to generate proposals for PASCAL VOC and COCO respectively.

Table 3: **Ablation study of our method.**

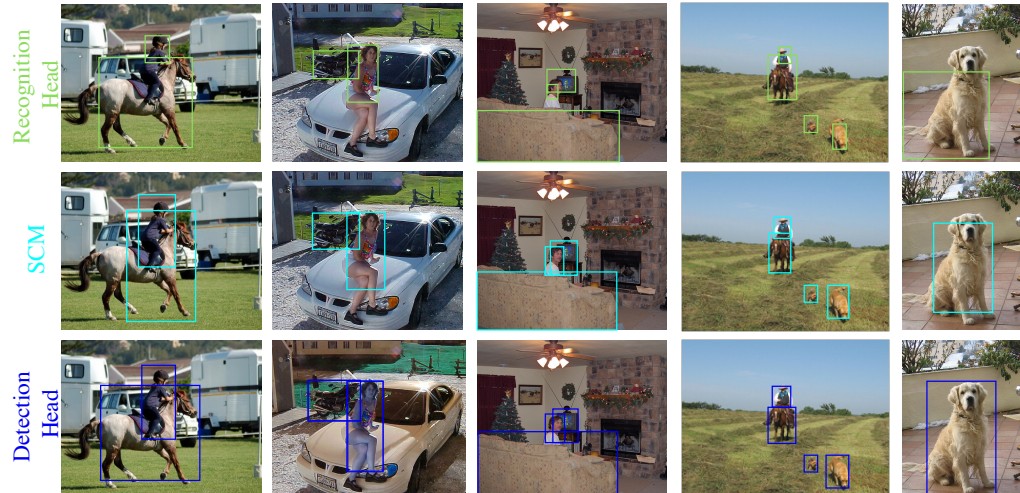

Figure 4: **Detection results on *novel* objects.** The results are from our proposed model but with different heads. The first row shows the results of the recognition head. The second row lists the results from SCM. The third row displays the results from the detection head.

## 5.2 COCO

**Setup.** We train on the COCO *train2017* split and test on *val2017* split. We simulate the cross-supervised object detection scenario on COCO by splitting the 80 classes into base and novel classes. We use a 20/60 split same as Hu et al. (2018), dividing the COCO categories into all the 20 classes contained in PASCAL VOC and the 60 that are not. We refer to these as the 'voc' and 'non-voc' category sets. 'voc→non-voc' indicates that we take 'voc' as our base classes and 'non-voc' as our novel classes. Similarly, we split the first 20 classes into 'twenty' and the last 60 classes into 'sixty'.

**Implementation details.** The implementation details are the same as § 5.1 by default. We train the whole framework for 13 epochs. There is no supervision from recognition head to detection head in the first 5 epochs. The learning rate is reduced by a factor of 10 at $8^{th}$, and $12^{th}$ epochs.

**Baselines and competitors.** Most baselines and competitors are the same as § 5.1. 'Rec. Head' represents only using our recognition head structure as a weakly supervised object detector.

**Results.** The results on COCO still support our discussion in § 5.1. Even in complex multi objects scenes, our DRN outperforms all baselines and competitors by a large margin.

## 5.3 ABLATION EXPERIMENTS

**Heatmap synthesis.** In Table 3a, we compare the different methods to synthesize the heatmaps in the spatial correlation module. For each position in the heatmap, we consider three kinds of values: the maximum of confidence, the sum of confidence, and the number of proposals covering the position. This result informs us to use max and sum to create a two-channel heatmap.

**Structure of SCM.** In Table 3b, we compare different implementations of SCM. We compare the FCOS (Tian et al. (2019)) with 5 convolutional layers and the standard FCOS with a ResNet-50 (He et al. (2016)) backbone. We also compare to the regression baseline mentioned in § 4. Considering the computation cost, we choose FCOS with 5 convolutional layer as our heatmap detector.

**Structure of the Recognition head.** In Table 3c, we compare different structures for the recognition head. WSDDN (Bilen & Vedaldi (2016)) and OICR are compared to our structure. The results support that our model can benefit from a stronger recognition head.

**Different proposal generation methods.** Table 3d shows the ablation of different ways to generate proposals. In PASCAL VOC with only 10 base classes, RPN performs worse than selective search. In COCO with 60 base classes, RPN performs better than selective search.

**Visualization.** Fig. 4 shows detection results on novel objects. Images in the first row, the second row, and the third row are detected by our model from the recognition head, the SCM, and the detection head respectively. The images in the first row tend to focus on the discriminating parts of the objects, e.g. the first and the second images contain only a part of the person. It also tends to detect co-occurring objects, e.g. the fourth image not only detects horse but also a large part of the person. Our SCM alleviates these problems. It tends to focus on the whole object, e.g. the first and the third samples detect the whole person instead of only the head. Also, it can correct unsatisfactory bounding boxes distracted by co-occurring objects, e.g. SCM correctly localizes the horse instead of localizing both the person and the horse in the fourth example. Obviously, bounding boxes in the third row are the best, indicating the efficacy of our framework.

## 6 CONCLUSION

In this paper, we have focused on cross-supervised object detection in realistic settings with complex imagery. We explore two major ways to build a good cross-supervised object detector: sharing network backbone between a recognition head and a detection head, and learning a spatial correlation module to bridge the gap between recognition and detection. Significant improvement on PASCAL VOC and COCO suggests a novel and promising approach for expanding object detection to a much larger number of categories.

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
