# OpenReview forum: "CROSS-SUPERVISED OBJECT DETECTION"
_ICLR.cc/2021/Conference — Reject_

### Official Review · AnonReviewer4 · 2020-10-20
**Good paper in every aspect, but with limited novelty.**

**Rating:** 6
**Confidence:** 4

**Review:**

This paper introduces a new method for training an object detector on a dataset that consists of some object categories with instance-level bounding box annotations, as well as some other object categories with only image-level labels. The topic is interesting, important, and potentially very useful for real applications. The authors propose an idea to transfer knowledge from a weakly supervised (WS) detection head into a fully supervised (FS) detection head, by producing pseudo-ground-truth bounding boxes for classes with image-level labels. The idea is straightforward and interesting. Experiments show significant and consistent gains in various scenarios. The paper is well-written.

The main drawback of this paper is the lack of substantial novelty. The authors claim to propose a "novel paradigm" named cross-supervised object detection, while this task has already been studied as reviewed in Section 2. Particularly, the idea of jointly training WS and FS heads on WS and FS labels with a shared backbone has been explored before (e.g. in [1], which should have been discussed in the related work). The idea of using a WS model to create pseudo-ground-truth bounding boxes for training another model has also been studied before (e.g. in [2]). The only novel idea of this paper seems to be the Spatial Correlation Module (SCM), which is discussed more below.

SCM is used to transform proposal boxes selected by the WS detector head into more accurate bounding boxes to serve as pseudo-ground-truth for WS classes. To this end, the authors train a class-agnostic bounding box refinement module on FS classes, and apply it on the proposals of WS classes. However, a similar result could have been (probably) achieved by simply replacing the bounding box regression head of the Faster R-CNN with a class-agnostic head, and training it on base classes, while using it to refine the proposals of both base and novel classes during test. The authors did not explore such simple alternatives to SCM.

Another issue is that although the authors cited several existing methods for cross-supervised object detection in Section 2, they did not discuss why the proposed method is superior, and they did not include them in performance tables, claiming "these methods can only perform object localization in single object scenes." I cannot verify the correctness of this claim, as any proposal-based model can detect multiple objects per image and per class.

Nevertheless, the paper has a clear motivation and idea, a scientifically sound analysis, and significant results and insights that can be helpful for future work. Therefore, I recommend acceptance. If the authors can convince me that the paper also has substantial novelty and advantages over all existing works, I am willing to raise my rating.

Minor comment: In equation (1), the two terms of the binary cross-entropy should probably be placed in parentheses, so the sum is applied on both terms. Also in the paragraph above eq (1), that loss term should be defined as a multi-label cross-entropy, rather than multi-class.

[1] Yang, Hao, Hao Wu, and Hao Chen. "Detecting 11k classes: Large scale object detection without fine-grained bounding boxes." Proceedings of the IEEE International Conference on Computer Vision. 2019.

[2] Tang, Peng, et al. "Multiple instance detection network with online instance classifier refinement." Proceedings of the IEEE Conference on Computer Vision and Pattern Recognition. 2017.

######## Post-Rebuttal Updates:

After reading the authors' response and other reviewers' opinions (especially R3), I would like to downgrade my rating slightly from 7 to 6. I still think the paper makes valuable contributions, but I also think the contributions are overstated and not precisely justified. Particularly:

1. I agree with R3 that the limitation of novelty should be considered from the two perspectives of "task" and "method". The task is certainly not new, which should be made clear in the paper. The newest version of the paper still claims "we define a new task—cross-supervised object detection"

2. The method is indeed somewhat new, due to the use of multi-task learning and SCM, but its novelty should be clarified, and compared to all similar methods, not only some of them which are weaker. For instance, the authors did not adequately justify whether/why their method is superior to YOLO 9000, or Yang et al ([1] above). They did mention some differences in response to R3's comments and mine, but I am not convinced. Moreover, the authors did not explicitly discuss and compare those distinctions in the paper, neither quantitatively nor intuitively.

3. In response to R3, The authors claim they "are the first to address this problem in situations of realistic complexity", which is not accurate. Particularly, the paper reads "While several works [...] have explored this problem before, [...] They struggle to learn under more complex and realistic scenarios, where there are multiple objects from potentially very different classes" This is not entirely true, as Yang et al. successfully evaluate on Open Images (and also VOC and COCO in their supplementary materials). The authors do not provide a convincing reason or evidence of existing methods "struggling" in realistic settings. YOLO 9000 is open-source, and could have been compared to the proposed method to confirm that claim.

Accordingly, I strongly encourage the authors to refine their claims and lay more emphasis on the actually novel aspects of their method, by thoroughly comparing those novelties with all similar methods. I still hope to see this paper accepted, but cannot endorse it due to insisting on inaccurate claims.

---

> ### Author Response · Authors · 2020-11-21
> **Response to Reviewer 4(2/2)**
>
> Q3:  Similar results could have been (probably) achieved by simply replacing the bounding box regression head of the Faster R-CNN with a class-agnostic head, and training it on base classes, while using it to refine the proposals of both base and novel classes during tests.
>
> A3: Good question. In Table 3(b), we provide an ablation study of using a class-agnostic 2/3/4 FC layer to regress and refine the proposal: “using three fully-connected layers to regress the ground truth location taking the coordinates of high-confidence bounding boxes as input”. However, this setting still has very minor differences from the suggestion of R4 (inputs are different). In our very initial experiments, we did exactly what R4 suggested. However, using a class-agnostic regression head learned on base classes actually decreases the performance compared to not regressing the predicted bounding boxes of novel classes. This is also why we designed our SCM.
>
> Q4: Lack of competitors.
>
> A4: In Table.1, we include two cross-supervised object detection competitors -- MSD (Zhang et al. 2018), and Weight Transfer (Kuen et al. 2019). The methods mentioned in related work, either follow a very different setting, (e.g., Gao et al. 2019 focus on semi-supervised object detection), either only report results on single object dataset ILSVRC2013 (e.g., Hoffman et al. 2014, Tang et al.).
>
> We also conducted experiments on COCO following Uijlings et al. 2018. Note that they have additional semantic information and their model learned from base classes in stage 1 and learned from novel classes in stage 2, whereas we learn from base and novel classes jointly. We use the same architecture and the same training and testing data. Our CorLoc for IOU>0.5 is 59.4 while theirs are 26.2.
>
> Q5: Why is the proposed method superior?
>
> A5:
> * One advantage is that joint training allows us to benefit from additional base-class labels. We are doing multi-task learning from two types of supervision and different classes may have different strengths of supervision, which has rarely been explored before.
> * The spatial correlation module allows us to transfer knowledge from base classes to novel classes.
> * Ultimately, the performance on more realistic data sets like COCO demonstrates that this works in more realistic settings. Other methods struggle to perform on these complex multi-object scenarios.
>
> Q6: Minor comment in equation(1).
>
> A6: Thanks! We will fix that.

---

> ### Author Response · Authors · 2020-11-21
> **Response to Reviewer 4(1/2)**
>
> We appreciate the valuable opinions from R4. We will try to address the novelty concern and highlight our advantages over other works.
>
>
> Q1: The idea of jointly training weakly-supervised and fully-supervised heads on weakly-supervised and fully-supervised labels with a shared backbone has been explored before (e.g. in [1], which should have been discussed in the related work).
>
>
> A1: Yes, we agree that the structure proposed in [1] has some similarity to our structure and we will add it in the related work. However, there are some differences.
>
> First, they only shared the Conv Layers before the ROI Pooling (see Fig.2 for details) whereas we share the Conv Layers, RoI Pooling layers, and FC Layers. Their sharing structure is closer to our “Two Branch*” Baseline as they both only shared the Conv Layers before the RoI Pooling layer. We know this difference may look subtle, but it actually has a major effect on accuracy. As shown in Table 1, Two Branch* (same as their methods to share backbone) achieves 44.1%mAP whereas Two Head* (same as our methods to share backbone) achieves 47.1% mAP. Note that the comparison is fair since the only difference between Two Branch* and Two Head* is mentioned above and “everything else is the same”.
>
> Second, even though both [1] and we have weakly-supervised and fully-supervised heads, our fully-supervised head can detect both weakly-supervised and fully-supervised classes whereas theirs can only detect fully-supervised classes. The purpose of their fully-supervised head is to improve the shared CNN backbone and thus learn a better weakly-supervised detector. In comparison, our fully-supervised head is expected to detect the novel objects in evaluation. Our weakly-supervised head serves as a tool to extract detection information from class labels and further improve our fully-supervised head. In short, in [1], the fully-supervised head is an auxiliary head and in our work, the weakly-supervised head is an auxiliary head. In Figure 4, we compare the quality of detection results between using the weakly-supervised and fully-supervised heads to detect novel instances.
>
> Q2: The idea of using a weakly-supervised model to create pseudo-ground-truth bounding boxes for training another model has also been studied before (e.g. in [2]).
>
> A2:
> Yes, as we claim “A majority of WSOD methods (Tang et al. (2017); Wan et al. (2019); Wei et al. (2018)) find that re-training a new detector taking the top-scoring bounding boxes from a weakly supervised object detector as ground truth marginally improves the performance”. Lots of WSOD methods have a “detection head” that learned from pseudo instance-level annotations generated from a “recognition head”.
>
> However, their “detection head” only learns from pseudo labels and it acts similarly to the “recognition head”. It is more like a slightly improved version of the recognition head. In comparison, our detection head learns from both noisy pseudo labels of novel classes and the ground truth of base classes. Our “detection head” is very powerful when detecting base objects and thus have a better prior knowledge even for novel classes. As a result, it is a true “detection head” rather than an updated version of the “recognition head”.
>
> These differences may look subtle, but we believe our significantly better performance is evidence that they have a major effect on accuracy. In Table 1, when the detection head does not learn from the recognition head and we predict novel class using the recognition head, the performance is 47.1% (Two Head*). Note that in this case the recognition head still benefits from the shared backbone and has a refinement head inside its structure. When we add the connection between the recognition head and the detection head, even without the spatial correlation module, the performance is 52.4% (Ours* w/o SCM).
>
> Finally, part of our contribution is that our architecture allows us to benefit substantially from additional labeled examples on the base classes. What’s more, our pseudo labels are refined by our spatial correlation module before supervising the detection head, whereas other methods do not have this process.

---

### Official Review · AnonReviewer2 · 2020-10-28
**Innovative formulation, but more comparisons are needed**

**Rating:** 6
**Confidence:** 3

**Review:**

**Summary and contributions**:
The paper presents a new task formulation for transferring knowledge for object detection from fully labeled classes to weakly labeled ones, with better results for complex scenes. So the authors propose to learn object detectors for novel classes (which were not seen at training time but specified apriori), based only on that object class label and the bounding boxes for other objects in the image (from the base classes). Compared with other works, the solution claims to be better on the localization aspect, focusing on the object as a whole, not only on its discriminative parts. Nevertheless, there are very few competitors taken into account. The model combines three components: a detection head and a recognition head, based on the same, unified backbone architecture, and a spatial correlation module that aligns the two heads. The authors test their solution, Cross-Supervised Object Detection, on PASCAL VOC and COCO (multi-object scenes).

**Strengths**:
- Proposing a model that succeeds in learning both novel and base classes at the same time, on two heads, extracting information from the weak labels (recognition head) and using it as supervision for the second head (detection head).
- The qualitative results look significantly better compared with single heads, containing more of the entire object, not only the important/discriminatory parts as the authors pointed out.
- FCOS with only 5 conv layers reaches a good enough performance.


**Weaknesses**:
* How does this work compare with other WSOD recent methods, e.g. C-MIL, [A, B, C] from the experimental results point of view, and the amount of supervision?
* The text/figure should be adjusted to better map one each other: Sec. 3.2: “The image and proposals are fed into several convolutional layers”, but in the figure, the proposals skip those layers.
* Section 3.2 is rather hard to read and understand (it has several inaccuracies in the formulas).

**Quality**:
The paper is technically sound.

**Clarity**:
The paper is mainly clearly written (except for Section 3.2 Recognition Head).

**Novelty**:
The components are not novel, but the proposed learning paradigm is novel.

**Significance of this work**:
The impact of the work is significant, enabling a new way (CSOD) of transferring knowledge between classes, using weak labels.

**Typos**:
- “corresponds to the respective element of the matrix...” the following formula is wrong (d should be replaced with r)
- what does the upper index R mean and why is it useful?
- matrices should be capital letters in bold
- “which is expresses”


While I have some doubts regarding the comparison with other works, I hope they will be clarified during the rebuttal.


[A] High-Quality Proposals for Weakly Supervised Object Detection, Cheng, G., Yang, J., Gao, D., Guo, L., & Han, J..Transactions on Image Processing 2020

[B] Instance-aware, Context-focused, and Memory-efficient Weakly Supervised Object Detection. Zhongzheng Ren, Zhiding Yu, Xiaodong Yang, Ming-Yu Liu, Yong Jae Lee, Alexander G. Schwing, Jan Kautz. CVPR 2020

[C] Mixed Supervised Object Detection with Robust Objectness Transfer, Yan Li, Junge Zhang, Kaiqi Huang, Jianguo Zhang. PAMI 2019

---

> ### Author Response · Authors · 2020-11-21
> **Response to Reviewer 2**
>
> We appreciate the valuable opinion from you. Please see our responses and clarifications for your questions below.
>
> Q1: How does this work compare with other WSOD recent methods, e.g. C-MIL, [A, B, C] from the experimental results point of view?
>
> A1: In Table.1, we actually compare our results with [C](MSD-VGG16, MSD-Ens, MSD-Ens+FRCN) on PASCAL VOC. We outperform it by 7.5% mAP. Note that [C] is conducted under exactly the same setting as ours. The best performance of C-MIL, [A], [B], and [C] on COCO is 11.4 AP and 24.3 AP_50 whereas our method achieves 13.9 AP and 36.2 AP_50.  On PASCAL VOC, the best performance of C-MIL, [A], [B], and [C] achieves 52.1 mAP on novel classes whereas we achieve 55.7 mAP.
>
> Q2: How does this work compare with other WSOD in the amount of supervision?
>
> A2: We have the same amount of supervision on novel classes, but on the base classes, we have bounding boxes and class labels whereas WSOD methods only have class labels. Note that all our results (Table 1,2) are reported on the novel classes. So in general, we have extra supervision on another set of classes that have no overlap with the testing classes.
>
> Q3: The text/figure should be adjusted to better map onto each other: Sec. 3.2: “The image and proposals are fed into several convolutional layers”, but in the figure, the proposals skip those layers.
>
> A3: Good points! Sorry for the confusion and we will update our figure to fix that.
>
> Q4: Section 3.2 is rather hard to read and understand.
>
> A4: Section 3.2 mainly introduces the prior work Online Instance Classifier Refinement (Tang et.al 2017). We will update and make it easier to understand.

---

### Official Review · AnonReviewer3 · 2020-10-28
**Duplicate task settings and limited novelty**

**Rating:** 4
**Confidence:** 5

**Review:**

Paper summary:
The paper proposes a new task cross-supervised object detection, which trains object detectors on the combination of base class images with instance-level annotations and novel class image with only image-level annotations. A network with a recognition head which is trained by image-level annotations and a detection head which is trained by instance-level annotations is proposed for the task. To generate instance-level annotations for novel class images with only image-level annotations, the paper proposes a spatial correlation module to generate pseudo gt boxes from high-confidence boxes. Results on PASCAL VOC and COCO show that the proposed method obtains very promising object detection results for novel classes.


Strengths:

+ The paper is well written.

+ Promising experimental results are obtained by the proposed method.

+ The proposed spatial correlation module is interesting.


Weaknesses:

- Duplicate task settings.
The proposed new task, cross-supervised object detection, is almost the same as the task defined in (Hoffman et al. 2014, Tang et al. 2016, Uijlings et al. 2018). Both of these previous works study the task of training object detectors on the combination of base class images with instance-level annotations and novel class image with only image-level annotations. The work (Uijlings et al. 2018) also conducts experiments on COCO which contains multi-objects in images.
In addition, the work  (Khandelwal et al. 2020) unifies the setting of training object detectors on the combination of fully-labeled data and weakly-labeled data, and conducts experiments on multi-object datasets PASCAL VOC and COCO. The task proposed by this paper could be treated as a special case of the task studied in (Khandelwal et al. 2020).
We should avoid duplicate task settings.

- Limited novelty.
The novelty of the proposed method is limited.
Combining recognition head and detection head is not new in weakly supervised object detection. The weakly supervised object detection networks (Yang et al. 2019, Zeng et al. 2019) also generate pseudo instance-level annotations from recognition head to train detection head (i.e., head with bounding box classification and regression) for weakly-labeled data.


Review summary:
In summary, I would like to give a rejection to this paper due to the duplicate task settings and limited novelty.

Khandelwal et al., Weakly-supervised Any-shot Object Detection, 2020

---------- Post rebuttal ----------

After discussions with authors and reading other reviews, I acknowledge the contribution that this paper advances the performance of cross-supervised object detection.

However, I would like to keep my original reject score. The reasons are as follows.

Extending datasets from PASCAL VOC to COCO is not a significant change comparing to previous tasks. The general object detection papers also evaluated on PASCAL VOC only about five years ago and now evaluate mainly on COCO. With the development of computer vision techniques, it is natural to try more challenging datasets. So although this paper claims that this paper focuses on more challenging datasets, there is no significant difference between the tasks studied in previous works like [a] and this paper.

In addition, apart from ImageNet, the work [b] also evaluates their method on the Open Images dataset which is even larger and more challenging than COCO. The difference between the tasks studied in [b] and this paper is only that, [b] adds a constraint that weakly-labeled classes have semantic correlations with fully-labeled classes and this paper doesn't. This difference is also minor.

Therefore, the task itself cannot be one of the main contributions of this paper (especially the most important contribution of this paper). I would like to suggest the authors change their title / introduction / main paper by 1) giving lower wights to the task parts 2) giving higher weights to intuitions of why previous works fail on challenging datasets like COCO and motivations of the proposed method.

[a] YOLO9000: Better, Faster, Stronger, In CVPR, 2017

[b] Detecting 11K Classes: Large Scale Object Detection without Fine-Grained Bounding Boxes, In ICCV, 2019

---

> ### Author Response · Authors · 2020-11-21
> **Response to Reviewer 3(2/2)**
>
>
> Q4: Khandelwal et al. 2020 unifies the setting of training object detectors on the combination of fully-labeled data and weakly-labeled data, and conducts experiments on multi-object datasets PASCAL VOC and COCO.
>
> A4: We want to highlight that Khandelwal et al. 2020 is a recently released arxiv paper and **had not been published as a peer-reviewed paper** at the time of submission.  The instructions on the ICLR website suggest that it is not essential to compare to non-peer-reviewed works. But we can still discuss and compare with them to address the concern of R3. We will add it to the related work.
>
> Their method “is divided into two stages: base training and fine-tuning”, “During base training, instances from D_base are used to obtain a detector / segmentation network”, and “In the fine-tuning phase, the network is fine-tuned on D_novel”. Their paradigm is closed to “transfer learning” and our paradigm is closed to “multi-task learning”(as mentioned in A2). We discussed the difference and the importance of our paradigm in A2.
>
> Also, their method uses additional semantic information (300-dimensional GloVe [34] vector embeddings) to calculate the similarity between base and novel classes.
>
> Despite the difference, we can compare their results and ours on PASCAL VOC using the same backbone (VGG-16) and the same training and testing data. We follow their class split setting and run our method. We achieve 66.7% mAP whereas they achieve 58.7% mAP.
>
>
> Q5: Combining a recognition head and a detection head is not new in weakly supervised object detection. The weakly supervised object detection networks (Yang et al. 2019, Zeng et al. 2019) also generate pseudo instance-level annotations from the recognition head to training the detection head (i.e., head with bounding box classification and regression) for weakly-labeled data.
>
> A5: Yes, as we claim “A majority of WSOD methods (Tang et al. (2017); Wan et al. (2019); Wei et al. (2018)) find that re-training a new detector taking the top-scoring bounding boxes from a weakly supervised object detector as ground truth marginally improve the performance”, lots of WSOD methods (including Yang et al. 2019, Zeng et al. 2019) have a “detection head” that learned from pseudo instance-level annotations generated from an “recognition head”.
>
> However, their “detection head” only learns from pseudo labels and it acts similarly to the “recognition head”. It is more like a slightly improved version of the recognition head. In comparison, our detection head learns from both noisy pseudo labels of novel classes and ground truth of base classes. Our “detection head” is very powerful when detecting base objects and thus have a better prior knowledge even for novel classes. As a result, it is a true “detection head” rather than an updated version of the “recognition head”.
>
> These differences are subtle, but we believe our significantly better performance is evidence that they have a major effect on accuracy. In Table 1, when the detection head does not learn from the recognition head and we predict novel class using the recognition head, the performance is 47.1% (Two Head*). Note that in this case the recognition head still benefits from the shared backbone and has a refinement head inside its structure. When we add the connection between the recognition head and the detection head, even without spatial correlation module, the performance is 52.4%(Ours* w/o SCM).
>
> Finally, part of our contribution is that our architecture allows us to benefit substantially from additional labeled examples on the base classes. What’s more, our pseudo labels are refined by our spatial correlation module before supervising the detection head, whereas other methods do not have this process.

---

> > ### Comment · AnonReviewer3 · 2020-11-23
> > **More discussions about the duplicate task setting and novelty**
> >
> > Thanks for clarifying the differences between the task studied in this paper and the tasks studied in Hoffman et al. 2014, Tang et al. 2016, Uijlings et al. 2018, and Khandelwal et al. 2020, and also thanks for clarifying the novelty.
> >
> >
> > ----- For task -----
> >
> > I agree with the authors that the tasks studied in Hoffman et al. 2014, Tang et al. 2016, Uijlings et al. 2018,  and Khandelwal et al. 2020 focus more on transfer learning and this paper focuses more on multi-task learning. However, this difference is minor, and it is easy to extend the methods by Hoffman et al. 2014, Tang et al. 2016, and Uijlings et al. 2018 to the multi-task learning setting, e.g., for Uijlings et al. 2018, they could train their final round of detector on the data with both base classes and novel classes.
> > In addition, the task studied in Khandelwal et al. 2020 could also be treated as multi-task learning because they show promising results on both base classes and novel classes.
> >
> > Even without considering the work by Khandelwal et al., there are also other works train object detectors on the combination of base class images with instance-level annotations and novel class images with only image-level annotations under the multi-task learning setting [a, b], and there are only minor differences between the task studied in this paper and the tasks studied in [a, b].
> >
> > Therefore, the proposed task has only minor differences comparing to the previous tasks and is not novel enough. To downgrade the claims about the novel task, the authors need to change their title, rewrite their abstract and introduction, and re-organize their method part. Considering that epic changes are needed to revise the paper, I would like to suggest the authors rewrite their paper and resubmit the paper to the next conference.
> >
> >
> > ----- For method -----
> >
> > The main differences between the method proposed in this paper and the methods proposed in (Yang et al. 2019, Zeng et al. 2019) are the proposed spatial correlation module and using real labels from fully labeled data. These methodological differences are also the main contributions of this paper. The authors should revise their paper to show the real contributions of their work.
> >
> > [a] YOLO9000: Better, Faster, Stronger, In CVPR, 2017
> >
> > [b] Detecting 11K Classes: Large Scale Object Detection without Fine-Grained Bounding Boxes, In ICCV, 2019

---

> > > ### Author Response · Authors · 2020-11-23
> > > **Response to R3**
> > >
> > > We thank R3 for the reply. Below is our response to the concern.
> > >
> > > Q1: I agree with the authors that the tasks studied in Hoffman et al. 2014, Tang et al. 2016, Uijlings et al. 2018, and Khandelwal et al. 2020 focus more on transfer learning and this paper focuses more on multi-task learning. However, this difference is minor.
> > >
> > > A1: The reason the machine learning community has developed a large variety of different learning paradigms, including transfer, multi-task, low-shot, semi-supervised and so on, is that there are **important differences in how models are trained under each paradigm to maximize performance**.
> > >
> > > Q2: Uijings et.al. 2018 and Khandelwal et al. 2020 could be modified to multi-task learning.
> > >
> > > A2: **In our previous reply A2 and A4, we already made a comparison of performance.** For results on COCO compare with Uijlings et al. 2018, “Our CorLoc for IOU>0.5 is 59.4 while theirs are 26.2”. For results on PASCAL VOC compare with Khandelwal et al. 2020, “We achieve 66.7% mAP whereas they achieve 58.7% mAP”.
> > > Of course, many previous methods could be modified, but the point is that they have not yet been modified. If it were easy to modify and get such a large gain in performance (from 26.2 to 59.4), the previous authors certainly could have done so. A similar argument, that it would be easy to modify previous methods to get large gains, could be made for most papers in computer vision. We do not believe this is a fair criticism. For example, the VGG deep network was a set of very simple modifications to ZF-net, and yet today it has 47,000 citations.
> > >
> > > Q3: There are also other works train object detectors on the combination of base class images with instance-level annotations and novel class images with only image-level annotations under the multi-task learning setting [a, b], and there are only minor differences between the task studied in this paper and the tasks studied in [a, b].
> > >
> > > A3:
> > >
> > > In [b], they conduct experiments on ImageNet, which is the same as the ILSVRC data set and we already **discussed it in our previous reply A1**. Also, they claim “our method is the first semi-supervised fine-grained detection framework that explicitly exploit semantic/visual correlations between coarse-grained detection and fine-grained classification data” and they assume a very high coarse-grained relation between detection and classification classes, which is a very different situation.
> > >
> > > In [a], still, their classification categories are ImageNet categories and the classification data is still object-centered and single object image. [b] claim that “YOLO 9000([a]) can also be viewed as a semi-supervised detection framework, but it is no more than a naive combination of detection and classification stream and only relies on the implicit shared feature learning from the network”. In comparison, we add connection between our recognition and detection head. Please refer to our previous reply A5 for the importance of this connection.
> > >
> > > Q4: The proposed task has only minor differences compared to the previous tasks and is not novel enough.
> > >
> > > **Summary** and A4:
> > > We disagree with this statement. R3 has proposed several similar papers. We go through each paper one by one and reply.
> > > The papers mentioned by R3 either focus on transfer learning paradigms (Uijlings et al. 2018, Khandelwal et al. 2020) or focus on very simplified settings (Hoffman et al. 2014, Tang et al. 2016, [a], [b]).
> > >
> > > We believe this is not a minor difference and we discuss these differences in our previous and current reply.
> > >
> > > Here is a brief summary of our views:
> > >
> > > * In A1, we clarify the difference between transfer learning and multi-task learning. We focus on multi-task learning.
> > > * In A2, we show the efficacy of our methods compared to methods from similar settings. We show substantial increases in performance.
> > > * In previous reply A1 and second paragraph in Introduction of our paper, we clarified our important contribution of conducting cross-supervised object detection under complex multi-objects scenes. The point is that we are the first to address this problem in situations of realistic complexity.
> > >
> > >
> > >
> > > We have revised the abstract and introduction to make it clear that we are not the first to work on cross-supervised object detection, but that we are the first to apply it to realistic settings with complex imagery. We add one line in the Abstract and a short paragraph after the original third paragraph of the Introduction. Other than the abstract and intro, there are only very minor changes to the remainder of the document (such as removing the word “new” from “new paradigm” in the conclusion). While R3 suggested that the changes to the document would be “epic”, we believe they are quite modest, and reflect all of the raised issues.

---

> ### Author Response · Authors · 2020-11-21
> **Response to Reviewer 3(1/2)**
>
> We appreciate the valuable opinion from AnonReviewer3. Please see our responses and clarifications for your questions below.
>
> Q1: The proposed new task, cross-supervised object detection, is almost the same as the task defined in (Hoffman et al. 2014, Tang et al. 2016).
>
> A1: We carefully read the papers (Hoffman et al. 2014, Tang et al. 2016) and we agree that they worked on cross-supervised object detection.  Thus, we will not claim that it is a completely new paradigm in our final paper. However, we do believe Cross Supervised Detection is a good name for this setting, and, we believe these early attempts were in very simplified settings. So, we maintain that we are the first to address this problem in complex multi-object settings.
>
> As we claim in the paper “Early WSOD work (Hoffman et al. (2014)) showed fair performance by directly applying recognition
> networks to object detection”. In related work introducing cross-supervised object detection methods, we write “Hoffman et al. (2014) and Tang et al. (2016) propose methods of adaptation for knowledge transfer from classification features to detection features”.
>
> “However, these weakly supervised detectors perform poorly at localization. Most WSOD experiments have been conducted on the ILSVRC (Russakovsky et al. (2015)) data set, in which images have only a single object”.
>
> We argue that “The simplicity of these data sets limits the number and types of distractors in an image, making localization substantially easier” and “Learning from only class labels, it is challenging to detect objects at different scales in an image that contains many distractors”. In comparison, we conduct experiments under “complex multi-object scenes, such as the COCO dataset”, which is part of our important contribution.
>
> In the revised version of the paper, we will further clarify that (Hoffman et al. 2014, Tang et al. 2016) have almost the same setting as us and we focus on this problem in more realistic and challenging scenarios.
>
> Q2:  (Uijlings et al. 2018) has a similar task and also conducts experiments on COCO which contains multi-objects in images.
>
> A2: It is true that there is some similarity between our cross-supervised object detection (CSOD) and the task in Uijlings et al. 2018. However, there is a significant difference between our CSOD and their setting.
>
> Consider the following two paradigms:
> 	Paradigm 1 (“Transfer learning”). Train a model on a set of base classes. Then, using only the trained model (and not the original training examples), adapt that model to a new set of classes.
> 	Paradigm 2 (“multi-task learning”). Given training data for a set of base classes and a set of novel classes, where the data for each set of classes may be different, train the best possible model for the novel classes.
>
> Uijlings et al. 2018 is using paradigm 1 and our CSOD belongs to paradigm 2. This is similar to the distinction between **“multi-task learning (our paradigm)”** and **“transfer learning (their paradigm)”**, where you adapt a model to a new set of classes or situations. We argue that cross-supervised detection is analogous to multi-task learning, where the learning occurs jointly. The fact that the ML community has already distinguished between multi-task learning and transfer learning attests to the important difference between these types of paradigms.
>
> Also, as we claim, “Uijlings et al. 2018 use a proposal generator trained on base classes to transfer knowledge by leveraging a MIL framework, organized in a semantic hierarchy.” Uijlings et al. 2018 have additional semantic information.
>
> CSOD “has similarities to both transfer learning and semi-supervised learning, since it transfers knowledge from base class to novel class and has more information about some instances than other instances. However, CSOD represents a distinct and novel paradigm for learning”. We will further clarify the difference in the later version.
>
> Q3: Uijlings et al. 2018 also conduct experiments on COCO which contains multi-objects in images.
>
> A3: We have discussed the difference between our setting and the setting of  Uijlings et al. 2018 in A2. Despite the difference, we still can compare our methods and theirs using the same backbone and same class split up (taking PASCAL VOC classes as base classes and other classes as novel classes) on COCO. Note that since the setting has differences the comparison is not absolutely fair (as discussed in A2) but it still gives us a reference to the efficiency of our work. Our CorLoc for IOU>0.5 is 59.4 while theirs are 26.2.

---

### Official Review · AnonReviewer1 · 2020-10-31
**The paper proposes a new problem setting for learning object detectors using weak supervision as well as a deep learning solution. However, the presentation and its relation to previous works should be improved.**

**Rating:** 6
**Confidence:** 5

**Review:**


This paper defines cross-supervised object detection which learns a detector from both image-level and instance-level annotations. It proposes a unified framework along with a spatial correlation module for the task. The spatial correlation module is used for transfer mapping information from base categories to novel categories. It conducts experiments on the PASCAL VOC dataset and COCO dataset, demonstrating the effectiveness.

Pros:
(1) The proposed spatial correlation module is a novel and effective transfer module.
(2) The ablation studies are relatively complete.

Cons:
(1) The structure of the proposed spatial correlation module should be described in more detail. What is the meaning of “replacing the backbone and feature pyramid network with five max-pooling layers.” in the heatmap detection part?
(2) In Table 1, are the experimental settings of those competitors such as MSD-VGG16, MSD-Ens, and Weight Transfer et al. exactly the same as those used in this paper?
(3) In Table 2, it seems like the method taking the non-VOC as the base classes while Hu et al. (2018) use the non-VOC as those classes without mask annotation. Can you tell me the reasons for this choice?
(4) Some similar problem settings are defined in [a] and [b]. The paper fails to compare the problem settings and justify the usefulness of the proposed problem setting in real application scenarios.

Overall evaluation: The major contribution of this paper comes from the spatial correlation module. However, I still have some doubts about the structure of this module. Since this task is first presented, I want to make sure that the comparison is as fair as possible.

[a] Weakly- and Semi-Supervised Fast Region-Based CNN for Object Detection. Journal of Computer Science and Technology (JCST) 34(6): 1269–1278 Nov. 2019.
[b] LSTD: A Low-Shot Transfer Detector for Object Detection, AAAI 2018

---

> ### Author Response · Authors · 2020-11-21
> **Response to Reviewer 1**
>
> We appreciate the valuable comments from you. Please see our responses and clarifications for your questions below.
>
> Q1: What is the meaning of “replacing the backbone and feature pyramid network with five max-pooling layers”?
>
> A1: In FCOS (Tian et al. 2019), they use a backbone network and a feature pyramid network to produce five feature maps with different dimensions. We use 5 max-pooling layers that take the image as input and output 5 feature maps of different dimensions that have the same shape as the feature maps in FCOS. The other part is the same as FCOS.
>
> Q2: In Table 1, are the experimental settings of those competitors such as MSD-VGG16, MSD-Ens, and Weight Transfer et al. exactly the same as those used in this paper?
>
> A2: The experimental settings of MSD-VGG16, MSD-Ens, MSD_Ens +FRCN are exactly the same as in our paper. For Weight Transfer, they have different settings and we reproduce their methods under our experimental setting.
>
> Q3:  In Table 2, it seems like the method is taking the non-VOC as the base classes while Hu et al. (2018) use the non-VOC as those classes without mask annotation. Can you tell me the reasons for this choice?
>
> A3: Hu et al. report two results. One takes non-VOC (60 classes) as base classes and VOC (20 classes) as novel classes. The other one takes VOC as base classes and non-VOC as novel classes. So we are actually following their first split up scheme. We choose this because we want to have a sufficient number of base classes so that our spatial correlation module can successfully learn the class-agnostic mapping from coarse bounding boxes predicted by the recognition head to the ground truth.
>
> Q4: Some similar problem settings are defined in [a] and [b]. The paper fails to compare the problem settings.
>
> A4:
>
>  [a] (Wang et al. 2019) focus on semi-supervised object detection. As we claim, cross-supervised object detection (CSOD) “has similarities to both transfer learning and semi-supervised learning, since it transfers knowledge from base classes to novel classes and has more information about some instances than other instances. However, CSOD represents a distinct and novel paradigm for learning.”
> The major difference between our cross-supervised object detection and semi-supervised object detection is that we have **zero instance-level labels** for new classes. Also, out-of-data classes (base classes) are never present in the training set of semi-supervised learning whereas we need to jointly learn from both base and novel classes.
>
> [b] (Chen et. al 2018) focus on a few-shot object detection setting. The major difference is that we have zero instance-level labels and we need to learn from only class labels for new classes whereas few-shot object detection has a few instance-level labels and does not contain any form of weakly-supervised learning.
>
> We will add [a] and [b] in our related work and further clarify the difference.
>
> Q5: Justify the usefulness of the proposed problem setting in real application scenarios.
>
> A5: As we claim, “CSOD could be a promising approach for expanding object detection to a much larger number of categories”. We have the same real application scenarios as weakly-supervised object detection, but importantly, “weakly supervised object detectors do not work well in complex multi-object scenes” while our CSOD works better in this situation. R4 also agrees that “The topic is interesting, important, and potentially very useful for real applications”.

---

### Decision · Program_Chairs · 2021-01-07
**Final Decision**

**Decision:**

Reject

**Comment:**

After the rebuttal phase, all scores are borderline (6) or negative (4). Among the most confident reviewers (confidence 5), one gives 6 and one gives 4. The reviewer with confidence 4 gives overall score 6 but states they cannot support the paper. There were several concerns about the novelty of the task and method, the challenge of the experimental settings, missing comparisons to recent prior work in the original paper, etc. While the reviewers see merit, the paper can benefit from another revision before being accepted, including to better position the novelty of its method and perhaps reduce claims of novelty of the task.